# A new model for the HPA axis explains dysregulation of stress hormones on the timescale of weeks

Omer Karin ⓘ, Moriya Raz, Avichai Tendler, Alon Bar, Yael Korem Kohanim, Tomer Milo & Uri Alon* ⓘ

## Abstract

Stress activates a complex network of hormones known as the hypothalamic–pituitary–adrenal (HPA) axis. The HPA axis is dysregulated in chronic stress and psychiatric disorders, but the origin of this dysregulation is unclear and cannot be explained by current HPA models. To address this, we developed a mathematical model for the HPA axis that incorporates changes in the total functional mass of the HPA hormone-secreting glands. The mass changes are caused by HPA hormones which act as growth factors for the glands in the axis. We find that the HPA axis shows the property of dynamical compensation, where gland masses adjust over weeks to buffer variation in physiological parameters. These mass changes explain the experimental findings on dysregulation of cortisol and ACTH dynamics in alcoholism, anorexia, and postpartum. Dysregulation occurs for a wide range of parameters and is exacerbated by impaired glucocorticoid receptor (GR) feedback, providing an explanation for the implication of GR in mood disorders. These findings suggest that gland-mass dynamics may play an important role in the pathophysiology of stress-related disorders.

**Keywords** dynamical compensation; exact adaptation; endocrine circuits; mathematical models of disease; systems medicine

**Subject Categories** Computational Biology; Signal Transduction

**Mol Syst Biol. (2020) 16: e9510**

## Introduction

A major hormone system that responds to stress is the HPA axis (Tsigos & Chrousos, 2002; Hosseinichimeh *et al*, 2015; Melmed *et al*, 2015; Zavala *et al*, 2019). Activation of the HPA axis results in the secretion of cortisol, which has receptors in almost all cell types, and exerts widespread effects on metabolism, immunity, and behavior, to help the organism cope with stress. The HPA axis is organized in a cascade of hormones (Fig 1A): physiological and psychological stresses cause secretion of CRH from the hypothalamus (H in Fig 1A). CRH causes the pituitary corticotroph cells to secrete ACTH, which in turn causes the adrenal cortex to secrete cortisol. Cortisol negatively feeds back on the secretion of the two upstream hormones.

The HPA axis is dysregulated in a wide range of physiological and pathological conditions. HPA dysregulation is measured by an assay of HPA function called the CRH test. In the CRH test, CRH is administered and cortisol and ACTH dynamics are measured for a few hours (Fig 1B). Major depression is associated with a blunted (reduced) ACTH response (Holsboer *et al*, 1984; Gold *et al*, 1986b; von Bardeleben & Holsboer, 1989) in the CRH test (Fig 1C), as well as with elevated baseline cortisol (Murphy, 1991). Elevated cortisol and blunted ACTH responses are also observed in other conditions that involve prolonged HPA axis activation (Fig 1D), including anorexia nervosa (Gold *et al*, 1986a), alcohol abuse disorder (von Bardeleben *et al*, 1989), and pregnancy (Magiakou *et al*, 1996).

In these conditions, one can see three stages of HPA axis dysregulation after HPA over-activation stops (e.g., by weight normalization after anorexia, cessation of alcohol consumption, and childbirth, respectively). The stages are defined in Fig 1D. The first stage occurs right after the over-activation stops. In this early withdrawal stage, ACTH is blunted and cortisol is high. In the second stage of intermediate withdrawal, which occurs 2–6 weeks after resolution of HPA over-activation, cortisol returns to baseline levels but ACTH remains blunted. This ACTH blunting may be causal for some of the clinical aspects of these conditions, since ACTH is co-regulated with beta-endorphin that modulates pain and mood (Guillemin *et al*, 1977; Vale *et al*, 1981; Marrazzi & Luby, 1986; Adinoff *et al*, 2005; Racz *et al*, 2008; Peciña *et al*, 2019). In the third stage, months after withdrawal, both ACTH and cortisol normalize (for alcohol abuse disorder this time-point was not measured in von Bardeleben *et al*, 1989; Fig 1D).

The dysregulation of the HPA axis is not explained by the existing mechanistic models of the HPA axis. The models cannot show the middle withdrawal phase with persistent blunting of ACTH responses despite the resolution of hypercortisolemia. This is because in current models, the dynamics of ACTH are strongly associated with the dynamics of cortisol, and so once cortisol normalizes, so should ACTH, within minutes to hours. Likewise, current

Department of Molecular Cell Biology, Weizmann Institute of Science, Rehovot, Israel
*Corresponding author. Tel: +972-8-934-4448; E-mail: uri.alon@weizmann.ac.il

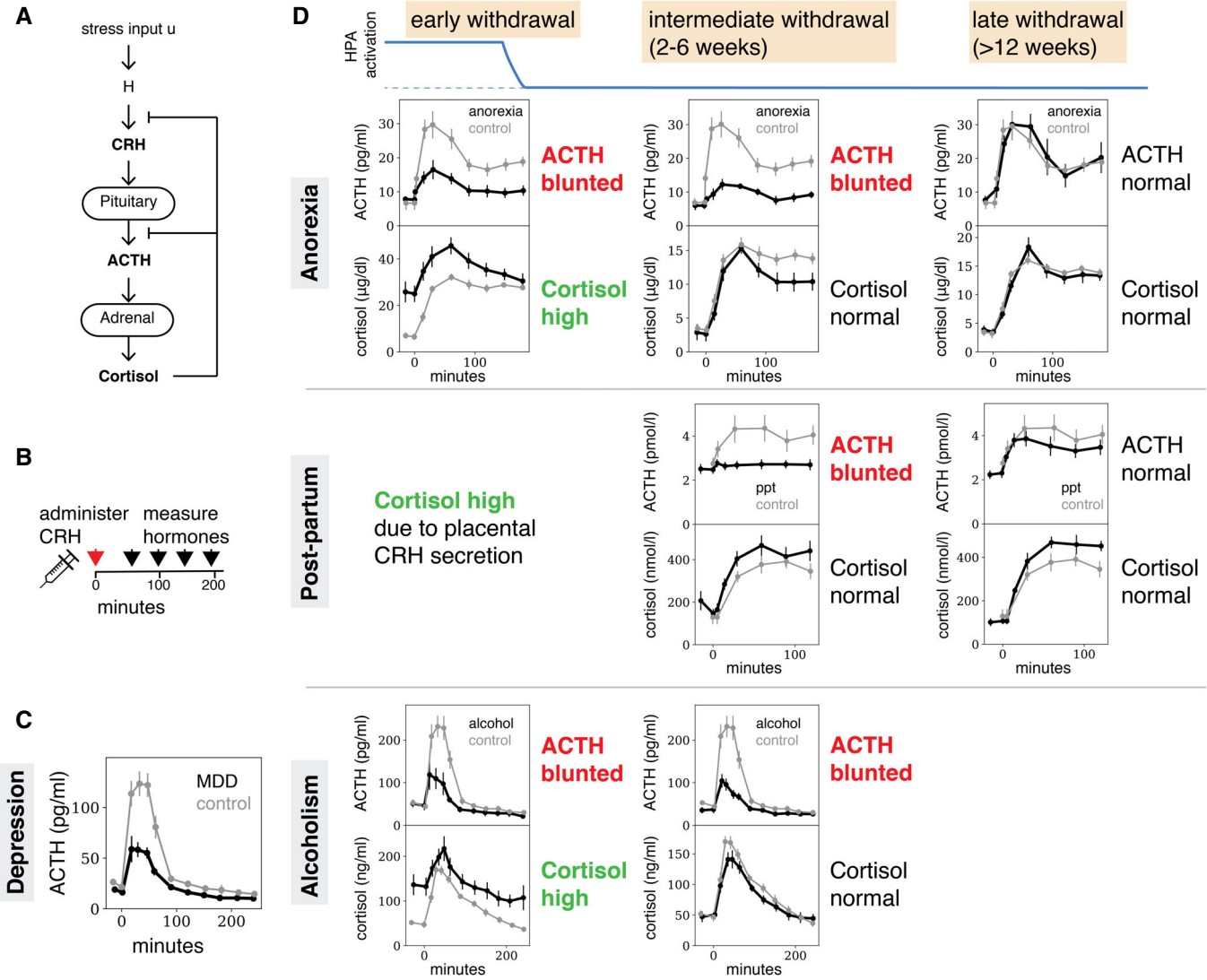

**Figure 1. After prolonged stress, ACTH response is blunted for weeks after cortisol response normalizes.**

A  Schema of the classic HPA axis. CRH causes the secretion of ACTH and cortisol.

B  In the CRH test, the secretion of these hormones is measured after CRH administration.

C  Patients suffering from major depressive disorder (MDD) show a blunted ACTH response to CRH (black line, *N* = 10), compared with controls (gray line, *N* = 10)—data from (von Bardeleben *et al*, 1988), shown are mean ± SEM.

D  Patients suffering from anorexia and admitted to treatment show a blunted ACTH response and hypercortisolemia, which resolves within 6–24 months after weight normalization—data from Gold *et al* (1986a). However, 3–4 weeks after weight normalization, cortisol dynamics are normal whereas ACTH dynamics are blunted. Pregnancy is associated with elevated cortisol levels due to CRH secretion by the placenta. 3 weeks after delivery, cortisol levels and dynamics return to normal, whereas ACTH dynamics are blunted—data from Magiakou *et al* (1996). After 12 weeks, ACTH dynamics normalize as well. Individuals recovering from alcohol abuse show hypercortisolemia and blunted ACTH response after admission—data from von Bardeleben *et al* (1989). After 2–6 weeks, these individuals show normal cortisol dynamics, but blunted ACTH responses persist. In all panels, control patient data are denoted by thin gray line (Anorexia: *N* = 13. Pregnancy: *N* was unspecified. Alcohol abuse disorder: *N* = 11), and case data by a thicker black line (Anorexia: left panel, *N* = 9, center panel, *N* = 5, right panel, *N* = 6. Pregnancy: *N* = 17. Alcohol abuse disorder: *N* = 20). Shown are mean ± SEM for all panels.

models cannot explain how a deficient ACTH response produces a normal cortisol response, given that ACTH is the main regulator of cortisol secretion. Explaining this dysregulation requires a process on the scale of weeks that decouples the dynamics of ACTH and cortisol. This timescale cannot be readily explained by existing models of the HPA axis, where the relevant timescale is the lifetime of hormones, which is minutes to hours (Bingzheng *et al*, 1990).

One important process with potentially a timescale of weeks is epigenetic regulation of the sensitivity of the cortisol receptor GR (glucocorticoid receptor) (Schaaf & Cidlowski, 2002; McGowan *et al*, 2009; Turner *et al*, 2010; Anacker *et al*, 2011; Cohen *et al*, 2012, 20; Watkeys *et al*, 2018). This process, however, cannot explain, on its own, the observed dysregulation. The reason is that GR resistance does not break the association between ACTH and

cortisol: GR resistance should cause both ACTH and cortisol levels to increase, in contrast to the observed ACTH blunting.

Here, we provide a mechanism for HPA axis dysregulation, and more generally for HPA dynamics on the time-scale of weeks. To do so, we add to the classic model two additional interactions which are experimentally characterized but have not been considered on the systems level. These are the interactions in which the HPA hormones act as the primary growth factors for the cells in their downstream glands. This causes the functional mass of the HPA glands to change over time, where by "functional mass of a gland" we mean the total capacity of the cells for the secretion of a hormone. A large body of research, beginning with Hans Selye in the 1930s, showed that the mass and number of adrenal cortisol-secreting cells increases under stress. Subsequent studies established the role of ACTH as the principle regulator of the functional mass of the adrenal cortex (Swann, 1940; Lotfi & de Mendonca, 2016). Imaging and postmortem studies also showed that adrenal mass increases in humans suffering from major depression (Amsterdam *et al*, 1987; Dorovini-Zis & Zis, 1987; Nemeroff *et al*, 1992; Szigethy *et al*, 1994; Rubin *et al*, 1996; Dumser *et al*, 1998; Ludescher *et al*, 2008) and returns to its original size after remission (Rubin *et al*, 1995).

Similarly, CRH causes the growth of pituitary corticotrophs that secrete ACTH. Prolonged administration of CRH, or a CRH-secreting tumor, leads to increases in corticotroph cell mass (Carey *et al*, 1984; Westlund *et al*, 1985; Schteingart *et al*, 1986; Gertz *et al*, 1987; Horvath, 1988; Asa *et al*, 1992; O'Brien *et al*, 1992) as well as ACTH output (Bruhn *et al*, 1984; Young & Akil, 1985). Adrenalectomy, which removes the negative feedback inhibition from the HPA axis, shows similar effects in rodents (Bruhn *et al*, 1984; Westlund *et al*, 1985; McNicol *et al*, 1988; Gulyas *et al*, 1991) and leads to increased proliferation of corticotrophs (Gulyas *et al*, 1991), which is potentiated by CRH treatment.

Changes in functional masses can occur by hypertrophy (enlarged cells) and/or hyperplasia (more cells); the exact mechanism does not matter for the present analysis. The changes in functional mass take weeks, due to the slow turnover time of cell mass. Such changes in functional mass have been shown in other hormonal axes (the insulin–glucose system) to provide important functions, including dynamical compensation, in which gland-mass changes buffer variations in physiological parameters (Topp *et al*, 2000; Ha *et al*, 2016; Karin *et al*, 2016).

We therefore asked whether the interplay of interactions between hormones and gland mass in the HPA axis can explain the observed dysregulation of the HPA axis on the timescale of weeks in the pathological and physiological situations mentioned above. We also tested other putative slow processes such as epigenetic regulation of GR (and more generally, GR resistance), slow changes in the input signal, or changes in the removal rate of cortisol.

Our model incorporates both the hormonal interactions and the gland-mass dynamics in the HPA axis. The model includes cortisol feedback through the high-affinity cortisol receptor MR (mineralocorticoid receptor) and the low-affinity receptor GR. We find that prolonged HPA activation enlarges the functional masses of the pituitary corticotrophs and adrenal cortex and that the recovery of these functional masses takes weeks after stress is removed. The dynamics of this recovery explains the observed HPA dysregulation: ACTH responses remain blunted for weeks after cortisol has normalized. Other putative slow processes that we tested cannot explain this dysregulation because they do not break the strong association between ACTH and cortisol. We further show that the GR protects the HPA axis against this dysregulation after high levels of stress, providing an explanation for the association between deficient GR feedback and depression. Finally, we demonstrate the physiological advantages conferred by the control of functional mass. Thus, functional mass changes provide an integrated explanation of HPA dysregulation and dynamics on the scale of weeks to months.

# Results

## A model for HPA axis dynamics that includes functional mass changes

We begin by showing that the classic HPA model cannot produce the observed dysregulation. We then add new equations for the gland masses and show that they are sufficient to explain the dysregulation.

The classical understanding of the HPA model is described by several minimal models (Gupta *et al*, 2007; Sriram *et al*, 2012; Andersen *et al*, 2013; Bangsgaard & Ottesen, 2017). These models are designed to address the timescale of hours to days and capture HPA dynamics on this timescale including circadian and ultradian rhythms. The input to these HPA models is the combined effects of physical and psychological stresses, including low blood glucose, low blood pressure, inflammation signals, psychological stressors, or effects of drugs such as alcohol. All inputs acting at a given time-point are considered as a combined input signal which we denote as $u$. The concentration of the three hormones CRH, ACTH, and cortisol are $x_1$, $x_2$, $x_3$. The three-hormone cascade, with feedback by $x_3$, is described by

$$\frac{dx_1}{dt} = k_1 g_1(x_3)u - w_1 x_1 \tag{1}$$

$$\frac{dx_2}{dt} = k_2 g_2(x_3)x_1 - w_2 x_2 \tag{2}$$

$$\frac{dx_3}{dt} = k_3 x_2 - w_3 x_3 \tag{3}$$

where the hormone secretion parameters are $k_1, k_2, k_3$, and hormone removal rates are $w_1, w_2, w_3$. Hormone half-lives, given by $\log 2/w_i$, are 4 minutes for $x_1$, 20 minutes for $x_2$, and 80 min for $x_3$ (Bingzheng *et al*, 1990; Table 1). The feedback functions $g_1(x_3)$ and $g_2(x_3)$ are Hill functions which describe the negative effect of cortisol on secretion of $x_1$ and $x_2$ (see Materials and Methods).

These equations have a single stable steady-state solution (Andersen *et al*, 2013). Their response to prolonged stress, namely a pulse of input $u(t)$ that lasts for a few weeks, shows elevated hormones during the stress, and a return to baseline within hours after the stress is over. Figure 2A shows a simulated CRH test, in which external CRH is added at a given time-point. ACTH shows no blunted ACTH responses in either early, intermediate, or late withdrawal phases (Fig 2A). This behavior can also be shown analytically (Materials and Methods).

**Table 1. Parameter values.**

| Parameter | Value |
| --- | --- |
| $w_1$ | 0.17/min (Andersen *et al*, 2013) |
| $w_2$ | 0.035/min (Andersen *et al*, 2013) |
| $w_3$ | 0.0086/min (Andersen *et al*, 2013) |
| $w_C$ | 0.099/day |
| $w_A$ | 0.049/day |
| $K_{GR}$ | 4 |
| $w_{CRH_E}$ | 0.016/min (Saphier *et al*, 1992) |
| $W$ | 30 min |
| $D$ | 20 |
| $w_{C_E}$ | 0.023/day |
| $\lambda$ | 1 |
| $w_R$ | 0.023/day |
| $n$ | 3 (Andersen *et al*, 2013) |

To describe HPA dynamics over weeks, we add to the classic model two interactions between the hormones and the total *functional mass* of the cells that secrete these hormones. We introduce two new variables, the functional mass of the corticotrophs, $C(t)$, and the functional mass of the adrenal cells that secrete cortisol, $A(t)$. To focus on the role of the mass, we separate the secretion parameters into a product of secretion per cell times the total cell mass. The secretion parameter of ACTH is thus $k_2 = b_2 C$, where $C$ is the corticotroph mass and $b_2$ is the rate of ACTH secretion per unit corticotroph mass. The parameter $b_2$ includes the metabolic capacity of the corticotrophs, the number of CRH receptors and the total blood volume which dilutes out ACTH. A similar equation describes the secretion parameter of cortisol, $k_3 = b_3 A$. To isolate the effects of mass changes, we assume for simplicity that the per-unit-biomass secretion rates $b_2$ and $b_3$ are constant, whereas $A(t)$ and $C(t)$ can vary with time.

This introduces two new equations for the functional masses, which have a slow timescale of weeks. Corticotrophs proliferate under control of $x_1$, and adrenal cortex cells under control of $x_2$, and thus

$$\frac{dC}{dt} = C(k_C x_1 - w_C) \tag{4}$$

$$\frac{dA}{dt} = A(k_A x_2 - w_A) \tag{5}$$

where the cell-mass production rates are $k_C x_1$ and $k_A x_2$, and the cell-mass removal rates are $w_C$ and $w_A$. The removal rates are taken from experimental data that indicate cell half-lives of days–weeks (Swann, 1940; Westlund *et al*, 1985; Gulyas *et al*, 1991). We use half-life of 6 days for $C$ and 12 days for $A$ (parameters given in Table 1), but the results do not depend sensitively on these parameters, as shown below.

The new model thus has five equations, three on the fast timescale of hours and two on the slow timescale of weeks. One

can prove that they have a single stable steady state. We non-dimensionalized the equations to provide hormone and cell-mass steady-state levels of 1 (Materials and Methods; equations 6–10).

**Model shows HPA axis dysregulation after prolonged activation**

The HPA model with gland-mass changes captures the experimentally observed dysregulation (Fig 2A and B). In response to a prolonged stress input $u$ of a few weeks or longer, the adrenal gland-mass A and corticotroph mass C both grow. When the prolonged stress input is over, the glands are large (Fig 2B—early withdrawal). They gradually return to baseline, but the corticotroph mass returns with an undershoot, dropping below baseline mass and then returning over several weeks (Fig 2B). These qualitative properties of the dynamics can be shown analytically (Materials and Methods) and do not depend on model parameters. These transients of gland masses are at the core of the hormone dysregulation.

The changed masses of the glands affect the response of the hormones to a CRH test. In early withdrawal, cortisol is high and ACTH responses are blunted (Fig 2B). Then, for a period of several weeks in intermediate withdrawal, cortisol has returned to its original baseline but ACTH remains blunted. Finally, after several months, both cortisol and ACTH return to their original baselines. Thus, the model recapitulates the experimentally observed dynamics of Fig 1.

To understand these dynamics in detail, we plot in Fig 3 the full behavior of the functional masses and hormones during and after the prolonged pulse of input (Fig 3A). Importantly, the qualitative conclusions are insensitive to the precise values of the model parameters. Figure 3 shows the dynamics of corticotroph and adrenal masses (Fig 3B), as well as the hormone levels (Fig 3C). To compare to CRH tests, we also simulated a CRH test at each time-point (Fig 3D). To visualize the result of the CRH tests, we plot the ratio of the peak hormone level after CRH administration relative to a control CRH test without the prolonged stress input (Fig 3E). Blunted responses correspond to values less than 1.

One can see two phases during the stress pulse. The initial phase occurs after the onset of the stressor and before adaptation to the stressor (Marked ONSET in Fig 3). It lasts several weeks. In this phase, the increase in input $u$ causes elevated levels and responses of CRH, ACTH, and cortisol (Fig 3C). However, over weeks of stress input, the corticotroph and adrenal masses grow (Fig 3B). These gland masses thus effectively adjust to the stressor, as example of the more general phenomenon of dynamical compensation in physiological systems (Karin *et al*, 2016). The mass growth causes a return to baseline of hypothalamic CRH and ACTH, due to negative feedback by cortisol. More precisely, a larger adrenal functional mass means that less ACTH is needed to produce a concentration of cortisol that drives ACTH down to baseline.

Such a return to baseline is called *exact adaptation*. Exact adaptation is a robust feature of this circuit due to a mathematical principle in the functional mass equations, equations (4 and 5), called integral feedback (Karin *et al*, 2016; Materials and Methods): The only steady-state solution of equations (4 and 5) is that the hormones $x_1$ and $x_2$ balance proliferation and growth parameters. Exact adaptation does not occur in models without the effects of functional mass changes—the hormones do not adapt to the stressor (Appendix Fig S1, Appendix Section 1).

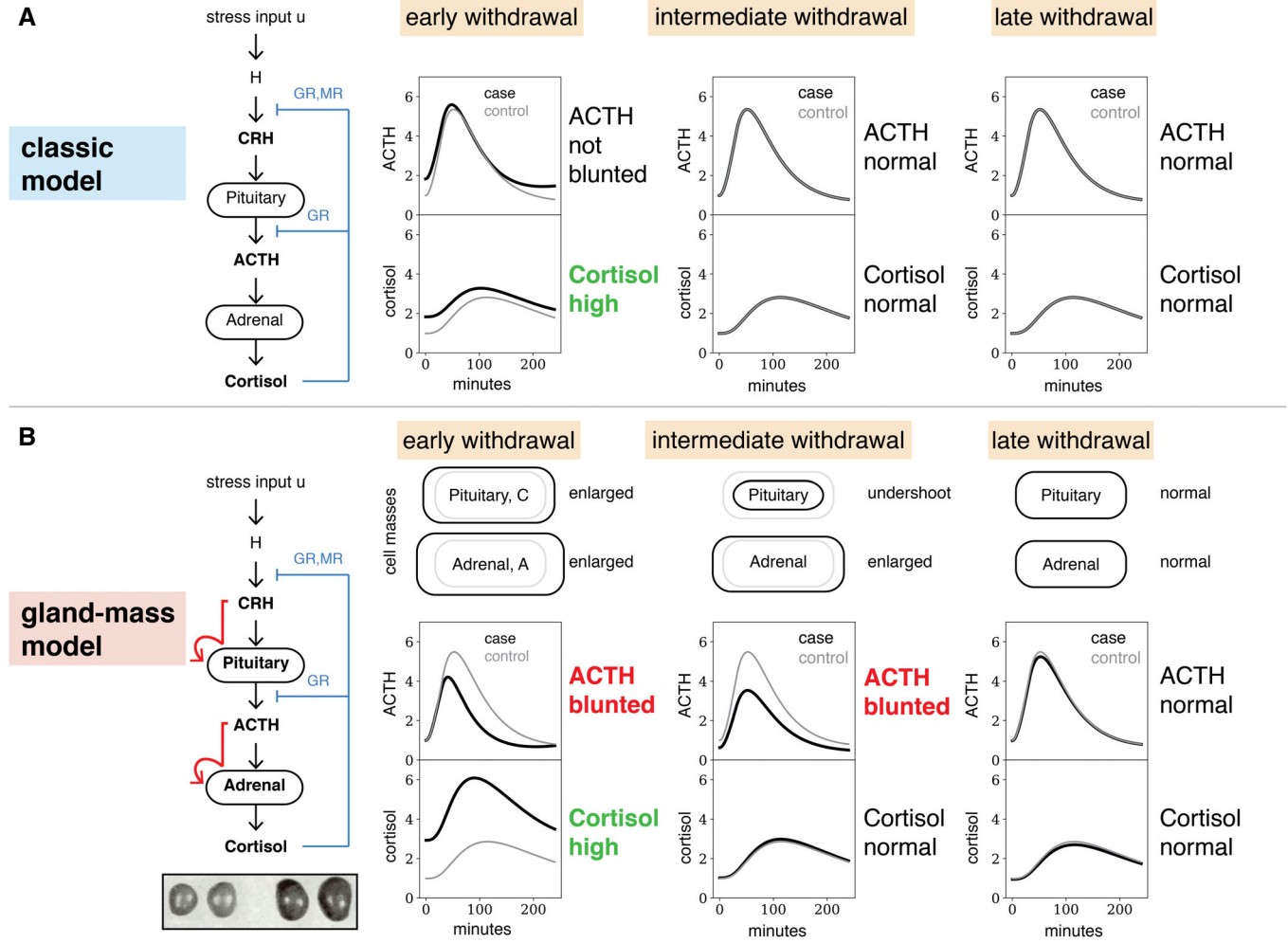

**Figure 2. Model with functional mass dynamics shows ACTH blunting for weeks even after cortisol normalizes.**

A The classic model of HPA axis dynamics without gland-mass dynamics produces elevated levels of stress hormones during prolonged stress. However, it does not produce blunted ACTH responses in the CRH test, and, after cessation of the stressor, all hormones return to baseline within hours.

B To account for the control of pituitary corticotroph growth by CRH and adrenal cortex growth by ACTH, we added to classic HPA model two equations that represent the dynamics of the functional mass of corticotrophs (C) and the adrenal cortex (A). (Inset) Such dynamics explain, for example, the enlarged adrenals of stressed rats (inset, right) compared with control (inset, left), adapted from Selye (1952). The model shows the three distinct phases of HPA axis dysregulation observed in experiments. At the end of a prolonged stress period, the adrenal mass is enlarged and the corticotroph mass is slightly enlarged, which results in hypercortisolemia and blunted ACTH responses in the CRH test. After a few weeks, corticotroph mass drops below baseline, while adrenal mass is slightly enlarged, causing normal cortisol dynamics with blunted ACTH responses to the CRH test. Finally, after a few months both tissue masses return to normal, leading to normalization of both cortisol and ACTH dynamics.

Data information: In all panels, simulations are of a CRH test (Materials and Methods), where "case" (black) is after stress and "control" (gray) is after basal HPA axis activation, as described in Fig 3.

The enlarged functional masses result in elevated cortisol levels during the stress period, but in adapted (that is, baseline) levels of CRH, ACTH, and blunted responses of CRH and ACTH to inputs (Fig 3E). During prolonged stress, there is thus a transition from an elevated to a blunted response of ACTH that occurs due to changes in functional masses, and results from cortisol negative feedback.

At the end of the prolonged stress pulse, the early withdrawal (or EW) phase, the functional masses are abnormal and take weeks to months to recover. This fundamental process is the reason for the hormonal dysregulation that is the subject of this study. In the first weeks after the stressor is removed, the adrenal and corticotroph functional masses shrink, accompanied by dropping cortisol levels. ACTH responses are blunted, and blunting may even worsen over time.

Then, cortisol and CRH levels and responses simultaneously normalize. This marks the beginning of the next phase, intermediate withdrawal (IW, Fig 3). In this phase, ACTH responses remain blunted, despite the fact that cortisol is back to baseline, because adrenal functional mass is enlarged, and corticotroph functional mass is deficient. Finally, over time, the entire dynamics of the HPA axis normalize (Late Withdrawal, LW in Fig 3), and the system has fully recovered.

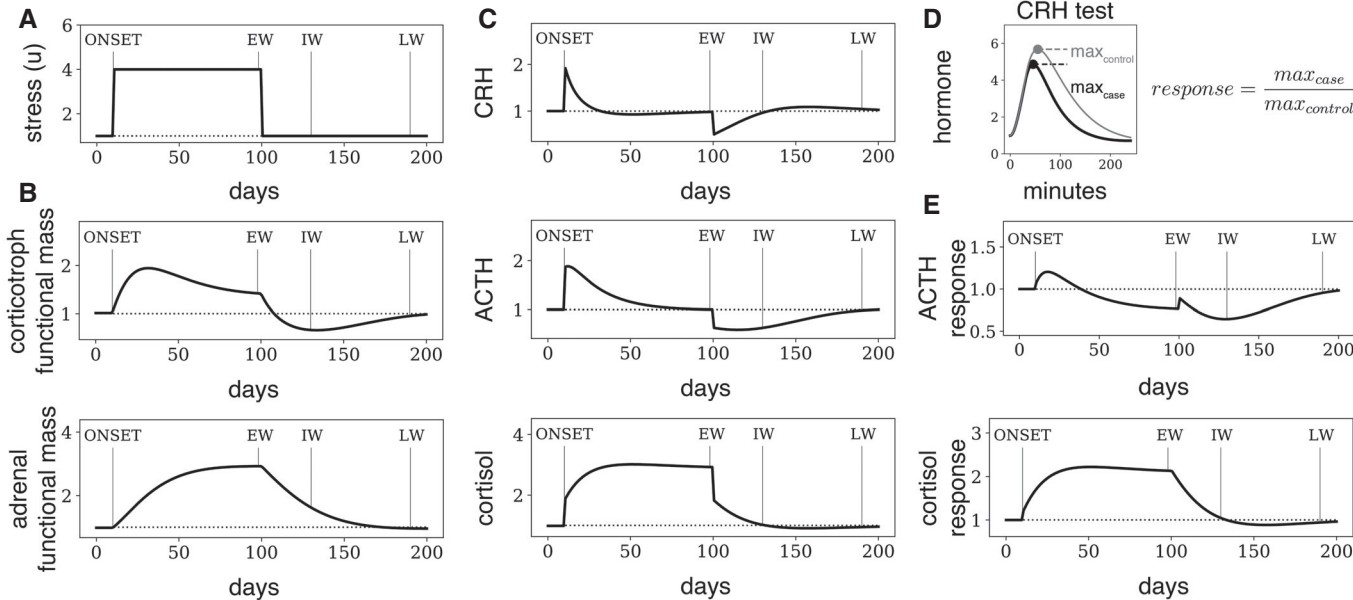

**Figure 3. Model dynamics of HPA axis after prolonged stress.**

A–C  Numerical solution of the HPA model after a prolonged pulse of input ($u$ = 4) lasting 3 months, followed by return to baseline input $u$ = 1 (A). During the pulse, gland masses (B) increase over weeks, leading to exact adaptation of ACTH and CRH levels after a few weeks (C) despite the increased input level. After stress ends, gland masses adjust back over weeks. During this adjustment period, the HPA axis is dysregulated.

D  To model a CRH test, we add exogenous CRH to the simulation and follow the hormones over several hours. The response is defined as the maximum response to a CRH test (case) relative to the maximum response to a CRH test in steady state with basal input conditions (control).

E  The response to a CRH test given at time $t$ is shown as a function of $t$. Thus, this is the predicted response to CRH tests done at different days during and after the stressor. The model (black lines) shows blunted (reduced) responses to CRH tests after the stress similar to those observed in Fig 1, as well as a mismatch between cortisol and ACTH dynamics that develops a few weeks after cessation of stress. EW is early withdrawal, IW is intermediate withdrawal and LW is late withdrawal.

These recovery phases are robust features of the HPA model with mass dynamics. After withdrawal of the stressor, cortisol and CRH levels and dynamics recover *together*, before the recovery of ACTH. This order occurs regardless of parameter values such as turnover times of the tissues (proof in Materials and Methods). The intuitive reason for this is that before CRH returns to baseline, the growth rate of the pituitary corticotrophs is negative, preventing ACTH from returning to baseline.

We conclude that the gland-mass model is sufficient to explain the dynamics of recovery from chronic HPA activation in the conditions mentioned in the introduction—anorexia, alcohol addiction, and pregnancy (Fig 3). In order to explain the timescales of recovery, the only model parameters that matter are the tissue turnover times. Good agreement is found with turnover times on the scale of 1–3 weeks for the corticotrophs and adrenal cortex cells (see Appendix Fig S2 for comparison of different turnover times, Appendix Section 2). The model therefore explains how ACTH responses remain blunted despite normalization of cortisol baseline and dynamics.

We also tested several alternative mechanisms with a slow time-scale of weeks. We tested models with constant gland masses and the following processes to which we assigned time constants on the order of 1 month: GR resistance following HPA activation (Appendix Fig S1, purple lines), slow changes in input signal (Appendix Fig S1, blue lines), and slow changes in cortisol removal rate (Appendix Fig S1, gray lines). None of these putative models

show the dysregulation that we consider. The reason is that these slow processes do not cause a mismatch between ACTH and cortisol needed to capture the blunting of ACTH despite normal cortisol responses during intermediate withdrawal.

## Deficient GR feedback exacerbates HPA dysregulation following prolonged stress

We next considered the role of glucocorticoid receptor (GR) in recovery from prolonged HPA activation. GR mediates the negative feedback of cortisol on CRH and ACTH secretion. Impaired feedback by GR is observed in many cases of depression. For example, administration of dexamethasone, which binds the GR in the pituitary, fails to suppress cortisol secretion in the majority of individuals suffering from depression (Coppen *et al*, 1983). Reduced expression of GR and impaired GR function in people with depression was also demonstrated in *post mortem* brains (López *et al*, 1998; Webster *et al*, 2002; McGowan *et al*, 2009; Pandey *et al*, 2013) and in peripheral tissues (Pariante, 2004). The feedback strength of the GR is regulated epigenetically and is affected by early-life adversity (Weaver *et al*, 2004; McGowan *et al*, 2009).

The relation between depression and impaired GR function seems paradoxical, since GR signaling mediates many of the detrimental effects associated with high cortisol levels such as hippocampal atrophy (Sapolsky *et al*, 1985). One explanation for the association between impaired GR feedback and depression is that

impaired GR feedback leads to failure of the HPA axis to terminate the stress response on the timescale of hours, leading to excessive cortisol levels (De Kloet *et al*, 2005; Anacker *et al*, 2011). GR feedback also plays a role in ultradian and circadian rhythms in the HPA axis (Stavreva *et al*, 2009; Walker *et al*, 2010; Sriram *et al*, 2012).

Here, we present an additional explanation for the association between GR feedback and depression, based on the functional mass dynamics model. Model simulations show that negative feedback by GR protects the HPA axis against large changes in hormone levels and responses on the timescale of weeks after a prolonged stress input (Fig 4A). The impaired dynamics of cortisol and ACTH is reduced when GR affinity is high. In other words, dysregulation is more severe the weaker the feedback from GR.

The reason for this effect is as follows. After an increase in stress levels ($u_1 \rightarrow u_2$), the adrenal gland mass increases (Fig 4B). When GR feedback is weak ($K_{GR} \gg u_2$), the adrenal increases by about a factor of $u_2/u_1$. However, if GR feedback is strong ($K_{GR} \ll u_2$), the adrenal increases to a smaller extent, because less cortisol is required to inhibit ACTH to the level required for precise adaptation. The smaller adrenal mass means a smaller dysregulation of cortisol and ACTH after stress. Strong GR feedback therefore provides resilience to the HPA axis against stress on the slow timescale.

We also tested the effects of variation in all other model parameters. We find that the hormone production and removal rates affect baseline levels but not the dynamics on the timescale of weeks. The parameters that do affect the timescale of weeks are the cell turnover times. We find that the present results are found for wide variations in these parameters (Appendix Fig S2, Appendix Section 2). We also tested other putative circuit designs in which the hormones control mass changes in alternative ways and find that the interactions considered here are among the very few designs that provide the observed HPA behavior (Appendix Section 3).

## Mass changes provide robustness and dynamic compensation to the HPA axis

Finally, we discuss potential advantages provided by the functional mass changes in the HPA axis. The first advantage is organ size control for the corticotrophs and adrenal cortex. The circuit provides a way for the cell populations to maintain their steady-state mass, by balancing growth and removal. Growth and removal must be precisely balanced in order to avoid aberrant growth or atrophy of the tissue. Thus, although the cells turn over, the feedback through the hormones couples with the hormone trophic effects to set the masses at a steady functional level.

The second beneficial feature is the robustness of the steady-state hormone levels with respect to physiological parameters. The simple form of the equations for the masses $C(t)$ and $A(t)$ (equations 4 and 5) locks the hormones CRH and ACTH into a unique steady state determined only by the growth and removal parameters of the tissues. Thus $CRH_{st} = \frac{w_C}{k_C}$, and $ACTH_{st} = \frac{w_A}{k_A}$. This is

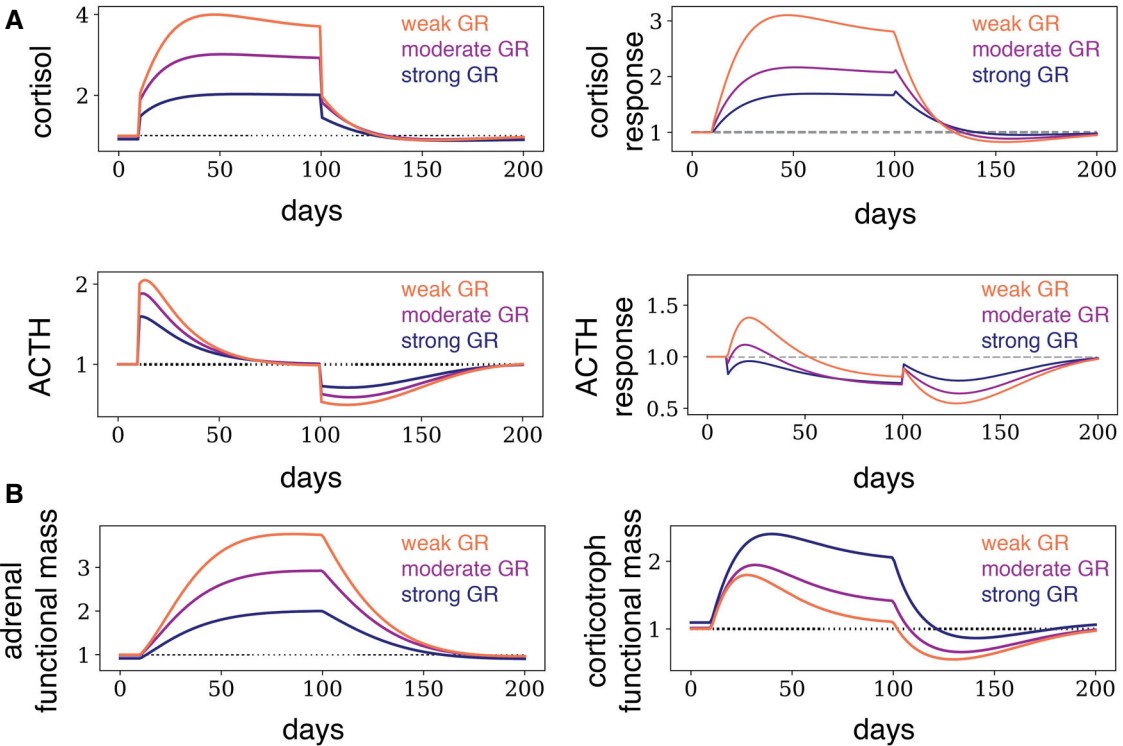

**Figure 4. GR provides resilience to the HPA axis against prolonged stressors.**

A, B   Here, we show the dynamics of the HPA axis during and after a prolonged pulse of stress, described in Fig 3, for $K_{GR} = 2$ (strong feedback), $K_{GR} = 4$ (moderate feedback), and $K_{GR} = 8$ (weak feedback). Stronger feedback from the GR attenuates the dysregulation of all HPA axis hormones.

remarkable because all other model parameters do not affect these steady states. Similarly, the cortisol baseline level, $\text{CORT}_{st} \approx \frac{k_1 k_C}{w_1 w_C} u$, depends on relatively few parameters. Cortisol baseline is proportional to the input $u$ to the hypothalamus (averaged over weeks), which corresponds to physiological and psychological stresses. Cortisol in the present analysis is the only HPA hormone that does not return to baseline after prolonged stress input. This may correspond to its physiological role as a stress response hormone, with multiple physiological endpoints, including modulation of immunity, metabolism, and behavior (McEwen, 1998). Cortisol therefore adjusts these endpoints according to averaged stress levels, allowing ongoing response as long as stress persists. Together with the persistent responsiveness of cortisol to stress input $u$, the circuit allows cortisol level to be independent on most other parameters, including the hormone production rates per unit biomass ($b_2, b_3$) or removal rates ($w_2, w_3$), as well as the rates of the proliferation and removal of the adrenal cortex cells, $k_A$ and $w_A$. This robustness is due to the ability of the functional masses to grow and shrink to buffer changes in these parameters. It allows cortisol to respond precisely to chronic stress input despite physiological variations in many of the circuit parameters.

This robustness also makes the steady-state level of cortisol and ACTH *independent of total blood volume*. This is because blood volume only enters through the production parameters $b_2$, $b_3$. These parameters describe the secretion rate per cell, and since hormone concentration is distributed throughout the circulation, these parameters are inversely proportional to blood volume [$k_1$ relates to the portal vein system that directly connects the hypothalamus and pituitary, and not to total blood volume (Owens & Nemeroff, 1991)]. The functional masses therefore adjust to stay proportional to blood volume.

The third feature of the present model is dynamical compensation, in which masses C and A change to make the fast-timescale (minute-hour) response of the system to acute stress inputs $u$ invariant to changes in the production rates $b_2$, $b_3$ as well as several other model parameters. A full analysis of dynamical compensation in the HPA axis is provided in the SI (Appendix Section 4).

# Discussion

In this study, we provide a mechanism to explain HPA axis dysregulation, based on the effects of the hormones on the gland masses. These effects are well characterized but have not been considered before on the systems level. We develop a minimal mathematical model that includes gland-mass changes and show that it is sufficient to explain the observed dysregulation following prolonged stress, alcohol abuse, anorexia, and postpartum.

The blunted ACTH response which we study may have clinical implications that last for weeks after stress is removed. One set of psychopathological implications is due to the fact that ACTH secretion is tightly linked with the secretion of the endogenous opioid β-endorphin. β-endorphin is cleaved from the same polypeptide as ACTH, the protein POMC (Guillemin et al, 1977; Rivier et al, 1982; Hargreaves et al, 1987) and is secreted in response to CRH from the anterior pituitary and from POMC neurons in the hypothalamus. Its secretion, like that of ACTH, is suppressed by cortisol (Lim et al, 1982). β-endorphin is the endogenous ligand of the mu-opioid

receptor (MOR), the primary target of addictive drugs such as morphine and heroin. It mediates euphoria and analgesia in humans and animals (Drewnowski et al, 1992; Hawkes, 1992; Peciña et al, 2006; Berridge & Kringelbach, 2008; Chelnokova et al, 2014; Buchel et al, 2018). Dysregulation of β-endorphin occurs in depression (Peciña et al, 2019), anorexia (Marrazzi & Luby, 1986; Marrazzi et al, 1990), and substance abuse disorders (Kiefer et al, 2002; Racz et al, 2008; Roth-Deri et al, 2008). Our results suggest that the dysregulation of β-endorphin, which we assume is similar to that of ACTH, persists for many weeks after cessation of stress. It can therefore contribute to pain and anhedonia in depression and addiction withdrawal.

A possible extension of the model is to incorporate hypothalamic neuropeptides that modulate the secretion of ACTH, such as arginine vasopressin (AVP). AVP is released in response to various stressors and acts synergistically with CRH to promote ACTH secretion. AVP receptors in the pituitary are upregulated in response to high cortisol levels (Aguilera & Rabadan-Diehl, 2000), which may help maintain responsiveness of ACTH secretion to some stimuli during chronic stress. Another relevant neuropeptide is oxytocin, which inhibits ACTH secretion (Legros, 2001). These neuropeptides, which have a half-life of a few minutes, are important for determining the acute response of the HPA axis to input stimuli. Therefore, modeling changes in their regulation on the slow timescale will help us better understand dysregulation after chronic HPA axis activation.

The present mass model provides insight into a physiological hallmark of several psychiatric disorders: deficient GR feedback. Impaired GR feedback is implicated in depression (De Kloet et al, 2005), and several genetic and environmental factors have been associated with this impaired feedback (Bet et al, 2009; Spijker & Van Rossum, 2009). Deficient GR feedback has been suggested to affect ultradian (Walker et al, 2010) and circadian (Sriram et al, 2012) HPA rhythms. We find that strong GR feedback (high GR affinity to cortisol) protects the HPA axis from dysregulation after prolonged activation. Reduced GR feedback causes larger dysregulation after prolonged stress. This effect is directly mediated by changes in adrenal and corticotroph masses: Strong feedback allows smaller changes of adrenal mass during the stress period, and hence to a smaller dysregulation after stress is removed. Since chronic stress can by itself lead to reduced GR sensitivity ("glucocorticoid resistance") (Cohen et al, 2012), the present findings suggest that prolonged stress makes the HPA less resilient to the next prolonged stress. This reduced resilience may be relevant to the progression of depressive episodes (Kendler et al, 2001).

The interactions between functional masses and hormones which underlie the present HPA axis dysregulation also provide exact adaptation to CRH and ACTH (Fig 3) during stress. Exact adaption has been extensively studied in the context of biochemical circuits (Ferrell, 2016; Alon, 2019), but less is known about the role of exact adaptation in physiological circuits, and, in particular, about its relevance to psychopathologies. One exception is mass changes in the insulin–glucose system, in which exact adaptation can provide control of glucose levels and dynamics (Karin et al, 2016) despite changes in insulin resistance, a form of dynamical compensation. It would be fascinating to study what functional roles exact adaptation has in the HPA axis, in order to understand the trade-offs that

impact its dysregulation. The present modeling approach may be able to address such questions.

We considered here changes in hormone production due to changes in total cell mass, rather than in intrinsic per-unit-biomass production rate parameters $b_2$ and $b_3$. These parameters could, in principle, also change as a function of time, although they may be limited by constraints on secretory capacity per unit biomass. Changes in production due to total cell-mass changes have been well documented in various endocrine systems, including the thyroid, the parathyroid, and pancreatic beta cells (Studer & Derwahl, 1995; Wada *et al*, 1997; Porat *et al*, 2011). Cell mass can grow by increasing cell numbers (hyperplasia) or by increasing cell size (hypertrophy). In the case of the HPA axis, both mechanisms occur—mice that were implanted with CRH-secreting tumors showed both corticotroph hyperplasia and hypertrophy (Asa *et al*, 1992), and chronically stressed rats showed both adrenal hyperplasia and hypertrophy (Ulrich-Lai *et al*, 2006). The relative importance of each mechanism may be species-specific. A recent study on the growth of pancreatic acinar cells showed that mice relied mostly on hypertrophy while humans relied on hyperplasia (Anzi *et al*, 2018). Characterizing relative contributions of hyperplasia and hypertrophy may have important implications for different aspects of the circuit, such as cell-mass turnover rates, and rates of mutation accumulation in the tissues.

The model presented in this study extends the literature on the feedback circuits that control the mass of endocrine organs. Such models are important for understanding disease processes that result in dysregulation of tissue mass, such as type 2 diabetes, in which insulin secretion by pancreatic β-cells becomes deficient. Pioneering work by (Topp *et al*, 2000; see also De Gaetano *et al*, 2008) showed that β-cell death at high levels of glucose, known as glucotoxicity, can result in bistability, leading to long-term glucose dysregulation. This work was extended by a more biologically accurate model that could explain additional data including pathways to diabetes (Ha *et al*, 2016). This bistability may be a side effect of a protective mechanism, because glucotoxicity can eliminate mutant β-cells that mis-sense glucose levels and as a result hyper-proliferate and hyper-secrete insulin (Karin & Alon, 2017). Another possible protective mechanism is autoimmune surveillance, where auto-reactive immune cells eliminate mis-sensing mutants (Kohanim *et al*, 2020), at the risk of the development of autoimmune diseases like type 1 diabetes. These mechanisms may be relevant for understanding certain diseases of the HPA glands, such as Cushing's disease due to ACTH-secreting pituitary adenomas, or Addison's disease due to autoimmune adrenal destruction (Kohanim *et al*, 2020).

From a medical point of view, if this model is correct, functional masses are potential targets to address HPA dysregulation. Measuring these masses and their dynamics using imaging may provide clinically relevant information. Interventions that seek to normalize functional masses can potentially reduce the extent of dysregulation during and after prolonged stresses. One class of interventions may use control-engineering approaches to control the masses. Such a feedback controller can work by periodically measuring masses or hormones and administering HPA agonists or antagonist at computed doses in order to guide the HPA axis to desired functional masses and hormone level goals (Ben-Zvi *et al*, 2009).

# Materials and Methods

## HPA axis model

We model HPA axis using the following non-dimensionalized equations, in which hormone and cell-mass steady states are equal to 1:

$$\frac{dx_1}{dt} = w_1(g_1(x_3)u - x_1) \tag{6}$$

$$\frac{dx_2}{dt} = w_2(Cg_2(x_3)x_1 - x_2) \tag{7}$$

$$\frac{dx_3}{dt} = w_3(Ax_2 - x_3) \tag{8}$$

$$\frac{dC}{dt} = w_C C(x_1 - 1) \tag{9}$$

$$\frac{dA}{dt} = w_A A(x_2 - 1) \tag{10}$$

The cortisol feedback on CRH is through the GR and MR receptors, and on ACTH only through the GR receptor. We model the effect of the high-affinity MR receptor, $M(x_3)$, as a Michaelis–Menten function in the saturated regime, because cortisol is thought to saturate MR at physiological levels (Andersen *et al*, 2013). Thus, $M(x_3) = 1/x_3$, where the Michaelis–Menten constant is absorbed in the production term parameters. We model the cooperative, low-affinity GR receptor using a Hill function $G(x_3) = 1/\left(1 + \left(\frac{x_3}{K_{GR}}\right)^n\right)$ with $n = 3$, where $K_{GR}$ is GR receptor halfway-effect parameter ($K_{GR} > > 1$). We model the combined receptor effects multiplicatively so that $g_1(x_3) = M(x_3)G(x_3)$, and $g_2(x_3) = G(x_3)$. All parameters are provided in Table 1. The quasi-steady state of equations (6–8) can be solved by setting time derivatives equal to zero, yielding at basal input $u = 1$:

$$x_2 \approx C^{1/2}A^{-1/2} \tag{11}$$

$$x_3 \approx (CA)^{1/2} \tag{12}$$

$$x_1 \approx (CA)^{-1/2} \tag{13}$$

## Alternative models

The model without functional mass dynamics is provided by equations (1–3). To this model, we added, instead of gland-mass dynamics, several alternative biological processes that have a slow timescale, potentially on the order of a month. The first process is glucocorticoid resistance, where chronically elevated cortisol levels cause weaker feedback from the GR (Schaaf & Cidlowski, 2002; Cohen *et al*, 2012). One possible mechanism involves epigenetic effects such as DNA methylation (Watkeys *et al*, 2018). To model GR resistance, we added a variable R that modifies the effective binding coefficient, yielding:

$$G(x_3) = \frac{1}{1 + \left(R\frac{x_3}{K_{GR}}\right)^n} \tag{14}$$

For $R$, we use the following equation, based on a model of leptin resistance by Jacquier $et\ al$ (2015). It describes the decline of $R$ with cortisol levels:

$$\frac{dR}{dt} = w_R(f(x_3) - h(x_3)R) \tag{15}$$

For high cortisol levels to induce strong resistance, we use the simple forms: $f(x) = 1$, $h(x) = 1 + \lambda x^2$.

The second alternative slow process is a slow decrease in input, $u$. We considered an exponentially decreasing input signal (Appendix Fig S1). The third alternative process is a putative slow decrease in cortisol clearance rate. To model this, we added a term $C_R$ to the removal term in equation (8):

$$\frac{dx_3}{dt} = w_3\left(A \cdot x_2 - x_3 C_R^{-1}\right) \tag{16}$$

Because there is no well-characterized biological process that governs removal on the scale of weeks, we use a putative description in which cortisol reduces its own removal rate by increasing $C_R$:

$$\frac{dC_R}{dt} = w_{C_R}(x_3 - C_R) \tag{17}$$

All putative processes were provided with a month timescale, by setting $w_{C_R} = w_R = \frac{\log(2)}{30}\text{day}^{-1}$. Simulation results are shown in the SI.

## CRH test

We modeled the CRH test by adding the following equation for the concentration of externally administrated CRH, denoted $x_{1E}$:

$$\frac{dx_{1E}}{dt} = w_{\text{CRH}_E}(\delta(t) - x_{1E}) \tag{18}$$

where $\delta(t)$ describes a dose $D$ of external CRH injected at $T_{inj}$ and lasting a time $W$:

$$\delta(t) = \begin{cases} D & T_{inj} < t \leq T_{inj} + W \\ 0 & \text{otherwise} \end{cases}$$

and the removal rate is $w_{\text{CRH}_E}$ as described in (Saphier $et\ al$, 1992). Extrinsically administrated CRH causes the pituitary to secrete ACTH, as described by adding $\text{CRH}_E$ to the intrinsic CRH (units of $\text{CRH}_E$ are set to have equal biological effect to CRH). Thus, equation (7) was modified to:

$$\frac{dx_2}{dt} = w_2(Cg_2(x_3)(x_1 + x_{1E}) - x_2) \tag{19}$$

The dose $D$ and pulse width $W$ were calibrated to provide the observed mean CRH test results in non-stressed control subjects, providing $D = 20$ and $W = 30$ min.

## Proof for dysregulation of ACTH after cortisol normalization

We briefly show how the model equations (6–10) provides ACTH dysregulation even after cortisol normalizes. Consider a prolonged stressor, like the one presented in Fig 3A (that is, a pulse increase in the input $u$). Since ACTH and CRH are adapted to the stressor, within hours after withdrawal of the stressor, CRH and ACTH levels drop to below their pre-stressor baseline, whereas cortisol remains above its pre-stressor baseline. CRH and cortisol recover simultaneously over a few weeks because CRH dynamics depend only on cortisol and the input $u$. It therefore remains to be shown that ACTH does not recover before cortisol. Since ACTH recovers from below baseline $x_2 < 1$, it recovers with a $positive\ temporal\ derivative$ $\frac{dx_2}{dt} > 0$. If (by negation) this happens before CRH and cortisol recover, we come to a contradiction: Since CRH is below its baseline $x_1 < 1$, $C$ has a negative derivative due to equation (9); the derivative of $A$ is zero due to equation (8). Because ACTH levels are approximately proportional to $C^{\frac{1}{2}}A^{-\frac{1}{2}}$ (equation 11), we find that ACTH has a $negative\ temporal\ derivative$ when it crosses its baseline $\frac{dx_2}{dt} < 0$ which is a contradiction. We conclude that CRH and cortisol return to baseline before ACTH does.

## Cortisol can show normal fast-timescale response despite ACTH blunting

We now use the HPA model equations (6–10) to show that when CRH and cortisol levels return to baseline after stress (point IW in Fig 3), their entire dynamics in response to any fast-timescale input (such as a CRH test) normalize, even if ACTH responses have not yet normalized. The mass of the adrenal cortex and pituitary corticotrophs at baseline is denoted $A_0, C_0$, and their size at the time-point IW where cortisol and CRH first normalize $(x_3 = x_1 = 1)$ is $\lambda_A A_0$, $\lambda_C C_0$. Because cortisol and CRH are at baseline, and both are functions of the product of gland masses $AC$ (equations 11–13) one obtains $\lambda_A \lambda_C = 1$. Replacing $\widetilde{x_2} = \lambda_C x_2$ in equations (6–8) yields revised equations for the fast-timescale dynamics:

$$\frac{dx_1}{dt} = w_1(u(t) \cdot g_1(x_3) - x_1) \tag{20}$$

$$\frac{d\widetilde{x_2}}{dt} = \frac{dx_2}{dt} \cdot \frac{1}{\lambda_C} = \frac{1}{\lambda_C}w_2(\lambda_C C_0 x_1 g_2(x_3) - \lambda_C \widetilde{x_2}) = w_2(C_0 x_1 g_2(x_3) - \widetilde{x_2}) \tag{21}$$

$$\frac{dx_3}{dt} = w_3(\lambda_A A_0 x_2 - x_3) = w_3(A_0 \widetilde{x_2} - x_3) \tag{22}$$

Note that equations (20 and 21) are the same as equations (6–8) when $A = A_0$ and $C = C_0$, that is, they are independent of $\lambda_A$, $\lambda_C$. In addition, the initial conditions for the hormones are also independent of $\lambda_A$, $\lambda_C$, because $x_3 = x_1 = 1$ are at baseline, while $x_2$, which is proportional to $C^{\frac{1}{2}}A^{-\frac{n_1}{1+n_1}} = C_0^{\frac{1}{2}}A_0^{-\frac{1}{2}} \cdot \lambda_C$, scales with $\lambda_C$, so that $\widetilde{x_2}_{ST}$ is also independent of $\lambda_A$, $\lambda_C$. We conclude that after cortisol and CRH return to baseline, the fast-timescale dynamics of cortisol and CRH to any input (such as a CRH test) are equal to the dynamics before the stressor. In the simulations, after CRH and cortisol have normalized for the first time, they only deviate from baseline

slightly (at most 9%), and so this consideration holds approximately for further times in the scenario of Fig 3.

### Software

All simulations were performed using Python 3.7.3.

## Data availability

Code to simulate the model and generate all the figures is available in the following database: https://github.com/omerka-weizmann/hpa_dynamics.

Expanded View for this article is available online.

### Acknowledgements

This project has received funding from the European Research Council (ERC) under the European Union's Horizon 2020 research and innovation program (grant agreement No 856487). UA is the incumbent of the Abisch-Frenkel Professorial Chair. OK is an Azrieli doctoral fellow.

### Author contributions

Conception and research performance: OK; UA. Methodology and modeling: OK; MR; AT; AB; YKK; TM; UA. Writing: OK and UA.

### Conflict of interest

The authors declare that they have no conflict of interest.

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
