## [Review Process File · Molecular Systems Biology]

A new model for the HPA axis explains dysregulation of stress hormones on the timescale of weeks

Omer Karin, Moriya Raz, Avichai Tendler, Alon Bar, Yael Korem Kohanim, Tomer Milo, and Uri Alon
DOI: [10.15252/msb.20209510](https://doi.org/10.15252/msb.20209510)

Corresponding author(s): Uri Alon (uri.alon@weizmann.ac.il)

Review Timeline:

Submission Date:	11th Feb 20
Editorial Decision:	29th Apr 20
Revision Received:	10th May 20
Editorial Decision:	7th Jun 20
Revision Received:	8th Jun 20
Accepted:	9th Jun 20

Editor: Jingyi Hou

Transaction Report:

29th Apr 2020

Manuscript Number: MSB-20-9510

Title: A new model for the HPA axis explains dysregulation of stress hormones on the timescale of weeks

Author: Omer Karin

Moriya Raz

Avichai Tandler

Alon Bar

Yael Korem Kohanim

Tomer Milo

Uri Alon

Dear Dr. Alon,

Thank you again for submitting your work to Molecular Systems Biology. One of the reviewers who initially accepted to review the manuscript finally dropped out and we were obliged to seek advice from a new expert. We have now heard back from the two reviewers who evaluated your manuscript. As you will see from the reports below, the reviewers think that the study is interesting and they acknowledge the quality and potential broader relevance of the presented data. However, they raise a series of -mostly minor- concerns, which should be carefully addressed in a revision of the manuscript. Since the reviewers' recommendations are rather clear, there is no need for me to reiterate all the points listed below.

On a more editorial level, please do the following:

- Please provide a .docx formatted version of the manuscript text (including legends for main figures, EV figures and tables). Please make sure that the changes are highlighted to be clearly visible.

- Please provide individual production quality figure files as .eps, .tif, .jpg (one file per figure).

- Please provide a .docx formatted letter INCLUDING the reviewers' reports and your detailed point-by-point responses to their comments. As part of the EMBO Press transparent editorial process, the point-by-point response is part of the Review Process File (RPF), which will be published alongside your paper.

- Please note that all corresponding authors are required to supply an ORCID ID for their name upon submission of a revised manuscript.

- We have replaced Supplementary Information with Expanded View (EV) Figures and Tables that are collapsible/expandable online (see examples in <http://msb.embopress.org/content/11/6/812>). A maximum of 5 EV Figures can be typeset. EV Figures should be cited as 'Figure EV1, Figure EV2' etc... in the text and their respective legends should be included in the main text after the legends of regular figures. Additional Tables/Datasets should be labeled and referred to as Table EV1, Dataset EV1, etc. Legends have to be provided in a separate tab in case of .xls files. Alternatively,

the legend can be supplied as a separate text file (README) and zipped together with the Table/Dataset file.

For the figures and tables that you do NOT wish to display as Expanded View figures/Tables, they should be bundled together with their legends in a single PDF file called *Appendix*, which should start with a short Table of Content. Appendix figures and tables should be referred to in the main text as: "Appendix Figure S1, Appendix Figure S2...Appendix Table S1" etc. See detailed instructions regarding expanded view here:

<https://www.embopress.org/page/journal/17444292/authorguide#expandedview>.

- Before submitting your revision, primary datasets (and computer code, where appropriate) produced in this study need to be deposited in an appropriate public database (see <https://www.embopress.org/page/journal/17444292/authorguide#dataavailability>).

The accession numbers and database should be listed in a formal "Data Availability " section (placed after Materials & Method) that follows the model below (see also <https://www.embopress.org/page/journal/17444292/authorguide#dataavailability>). Please note that the Data Availability Section is restricted to new primary data that are part of this study.

Data availability

- We would encourage you to include the source data for figure panels that show essential quantitative information. Additional information on source data and instruction on how to label the files are available at < <https://www.embopress.org/page/journal/17444292/authorguide#sourcedata> >.

- All Materials and Methods need to be described in the main text. We would encourage you to use 'Structured Methods', our new Materials and Methods format. According to this format, the Material and Methods section should include a Reagents and Tools Table (listing key reagents, experimental models, software and relevant equipment and including their sources and relevant identifiers) followed by a Methods and Protocols section in which we encourage the authors to describe their methods using a step-by-step protocol format with bullet points, to facilitate the adoption of the methodologies across labs. More information on how to adhere to this format as well as downloadable templates (.doc or .xls) for the Reagents and Tools Table can be found in our author guidelines: <

<https://www.embopress.org/page/journal/17444292/authorguide#researcharticleguide>>. An example of a Method paper with Structured Methods can be found here: .

- Please provide a "standfirst text" summarizing the study in one or two sentences (approximately

250 characters, including space), three to four "bullet points" highlighting the main findings and a "synopsis image" (550px width and max 400px height, jpeg format) to highlight the paper on our homepage.

When you resubmit your manuscript, please download our CHECKLIST (<http://bit.ly/EMBOPressAuthorChecklist>) and include the completed form in your submission. *Please note* that the Author Checklist will be published alongside the paper as part of the transparent process (<https://www.embopress.org/page/journal/17444292/authorguide#transparentprocess>).

If you feel you can satisfactorily deal with these points and those listed by the referees, you may wish to submit a revised version of your manuscript. Please attach a covering letter giving details of the way in which you have handled each of the points raised by the referees. A revised manuscript will be once again subject to review and you probably understand that we can give you no guarantee at this stage that the eventual outcome will be favorable.

Yours sincerely,

Jingyi Hou
Editor
Molecular Systems Biology

If you do choose to resubmit, please click on the link below to submit the revision online *within 90 days*.

Link Not Available

IMPORTANT: When you send your revision, we will require the following items:

1. the manuscript text in LaTeX, RTF or MS Word format
2. a letter with a detailed description of the changes made in response to the referees. Please specify clearly the exact places in the text (pages and paragraphs) where each change has been made in response to each specific comment given
3. three to four 'bullet points' highlighting the main findings of your study
4. a short 'blurb' text summarizing in two sentences the study (max. 250 characters)
5. a 'thumbnail image' (550px width and max 400px height, Illustrator, PowerPoint or jpeg format), which can be used as 'visual title' for the synopsis section of your paper.
6. Please include an author contributions statement after the Acknowledgements section (see <https://www.embopress.org/page/journal/17444292/authorguide>)
7. Please complete the CHECKLIST available at (<http://bit.ly/EMBOPressAuthorChecklist>). Please note that the Author Checklist will be published alongside the paper as part of the transparent process (<https://www.embopress.org/page/journal/17444292/authorguide#transparentprocess>).

8. Please note that corresponding authors are required to supply an ORCID ID for their name upon submission of a revised manuscript (EMBO Press signed a joint statement to encourage ORCID adoption). (<https://www.embopress.org/page/journal/17444292/authorguide#editorialprocess>)

Currently, our records indicate that the ORCID for your account is 0000-0003-1121-5907.

Link Not Available

The system will prompt you to fill in your funding and payment information. This will allow Wiley to send you a quote for the article processing charge (APC) in case of acceptance. This quote takes into account any reduction or fee waivers that you may be eligible for. Authors do not need to pay any fees before their manuscript is accepted and transferred to the publisher.

REFEREE REPORTS

Reviewer #1:

This manuscript develops a new model for the Hypothalamic-Pituitary-Adrenal (HPA) set of hormones. The hormones (except cortisol) return to normal baseline levels despite prolonged stress. However, temporal responses of all hormone levels after CRH injection (called the CRH test) differ from normal. Existing mathematical models do not incorporate the cell masses of glands producing these hormones and cannot reproduce the time courses following the CRH test at early and intermediate phases of recovery after stress withdrawal. The authors show that including the effective gland masses and their regulation in the model captures the experimentally observed behaviors, specifically the persistence of blunted ACTH response to the CRH test. Finally, the strength of Glucocorticoid Receptor (GR)-mediated negative feedback control in HPA resilience during prolonged stress and the benefits of mass changes are examined.

This is a clearly written, interesting manuscript explaining the workings of an important physiological system deregulated in many humans for various reasons. It also extends the concept of dynamic compensation from Karin et al. (2016) to a new physiological system, suggesting the generality of the concept. I would like to recommend publication once a few minor comments can be considered.

Minor points:

(1) There is a statement in the Introduction that epigenetic regulation of GR sensitivity cannot explain the CRH test results. Has this been shown in any of the references cited? If not then add a

reference to the Methods section showing epigenetic regulation is insufficient to capture the experimental data.

(2) Instead of cell masses, the production rate per each cell could increase to give the same outcome. That is, one could imagine $b_2(t)$ and $b_3(t)$ being time-dependent while C and A being constant. This does not happen, which probably suggests that there are some constraints on hormone output per individual cell. Can this be discussed a bit?

(3) Fig. 2b: when showing the cell masses, it may be useful to show the control cell masses somehow as well (preferably in gray). Also, label those rows "Cell masses".

(4) The cortisol baseline depends on very few parameters - but still depends on k -s and w -s. Also, this is the only hormone for which the baseline does not return to normal during prolonged stress. Why? In principle, could there be regulatory mechanisms to also push cortisol levels back to baseline? Why is it OK for human physiology to have high cortisol, but not OK to have abnormal levels of the other hormones?

(5) What would happen if both cell masses would depend on the same single hormone (either only x_1 or only x_2)?

Reviewer #3:

This is a very well written manuscript that suggests an explanation for data showing disconnection between ACTH secretion and CORT secretion following chronic stress. The beauty of the paper is the simplicity of the hypothesis: that everything can be explained by dynamic changes in the mass of the hormone-secreting cells. As demonstrated, if these changes occur on different slow time scales, then other features of the network (particularly negative feedback of cortisol) ensure that observations on hormone secretion levels that have been reported experimentally under a variety of stress-related conditions are replicated. No sensitivity analysis is needed, this is a very robust network response if one incorporates slow dynamic changes in cell mass. The work is a nice example of how a simple mathematical model can be used to explain ubiquitous data. I have only a few comments:

1) AVP is never mentioned. This is strange, since AVP is co-released with CRH from hypothalamic neurons and has a strong effect on corticotroph activity. I realize that much of the paper focusses on data from CRH tests (which don't involve AVP), but the physiological stimulatory role of AVP should at least be mentioned when the HPA axis is discussed.

2) Back in 2000 a model was published by Topp et al. (J. Theoretical Biology, 206:605) that demonstrated how beta-cell mass could change dynamically in response to insulin resistance so as to maintain glucose homeostasis. A later, and more biologically accurate, model was recently published by Ha et al. (Endocrinology, 157:624, 2016) that could explain other data including pathways to diabetes. Again, a main focus was on dynamic changes in the mass of the hormone-secreting (insulin in this case) cells. I get the feeling that the authors don't know about this work, which is based on dynamic compensation of cell mass (though I am not sure if that language was used). The authors only cite one of their own papers on the insulin-glucose system, which is also from 2016. Clearly, these other papers are relevant and should be cited and discussed as earlier models exhibiting dynamic compensation in an endocrine system.

3) The Fig. 1 caption incorrectly states that the thicker curves are control.

Dear Editor,

Thank you very much for the positive consideration of our manuscript and for the reviewer comments. We have now addressed all of the comments in the revised manuscript. The reviews helped us make the manuscript more clear and comprehensive. We detail below the point-by-point changes.

Yours

Uri

Reviewer #1:

This manuscript develops a new model for the Hypothalamic-Pituitary-Adrenal (HPA) set of hormones. The hormones (except cortisol) return to normal baseline levels despite prolonged stress. However, temporal responses of all hormone levels after CRH injection (called the CRH test) differ from normal. Existing mathematical models do not incorporate the cell masses of glands producing these hormones and cannot reproduce the time courses following the CRH test at early and intermediate phases of recovery after stress withdrawal. The authors show that including the effective gland masses and their regulation in the model captures the experimentally observed behaviors, specifically the persistence of blunted ACTH response to the CRH test. Finally, the strength of Glucocorticoid Receptor (GR)-mediated negative feedback control in HPA resilience during prolonged stress and the benefits of mass changes are examined.

This is a clearly written, interesting manuscript explaining the workings of an important physiological system deregulated in many humans for various reasons. It also extends the concept of dynamic compensation from Karin et al. (2016) to a new physiological system, suggesting the generality of the concept. I would like to recommend publication once a few minor comments can be considered.

We thank the reviewer for this endorsement.

Minor points:

(1) There is a statement in the Introduction that epigenetic regulation of GR sensitivity cannot explain the CRH test results. Has this been shown in any of the references cited? If not then add a reference to the Methods section showing epigenetic regulation is insufficient to capture the experimental data.

We thank the reviewer for this comment. To the best of our knowledge, it was not previously shown that regulation of GR sensitivity is insufficient to capture experimental data. However, we can show this by modelling the effects of GR sensitivity on ACTH and cortisol. We now revised the introduction and methods section to clarify this important point. In the introduction, we now write (page 6):

This process, however, cannot explain, on its own, the observed dysregulation. The reason is that GR resistance does not break the association between ACTH and cortisol: GR resistance should cause both ACTH and cortisol levels to increase, in contrast to the observed ACTH blunting.

We also revised the paragraph in appendix section 1 where we show that epigenetic regulation of GR sensitivity cannot explain the observed dysregulation:

A second alternative model is glucocorticoid resistance which depends on cortisol levels (Cohen *et al*, 2012; Merkulov *et al*, 2017). We model this mechanism by adding a cortisol-dependent decrease in the sensitivity of the GR (Methods, purple lines in Figure S1). This mechanism also does not result in blunted ACTH responses (in fact, it sensitizes ACTH responses), and there is no mismatch between cortisol and ACTH after recovery from prolonged stress. The reason for this is that varying GR sensitivity does not disrupt the correspondence between ACTH levels and responses to cortisol levels, so when ACTH responses are blunted, cortisol levels are low, and vice versa.

(2) Instead of cell masses, the production rate per each cell could increase to give the same outcome. That is, one could imagine $b_2(t)$ and $b_3(t)$ being time-dependent while C and A being constant. This does not happen, which probably suggests that there are some constraints on hormone output per individual cell. Can this be discussed a bit?

We thank the reviewer for this comment. We addressed this in a new discussion session, where we discuss hormone secretion per cell, as well as mechanisms for increase in cell mass (page 17):

We considered here changes in hormone production due to changes in total cell mass, rather than in intrinsic per-unit-biomass production rate parameters b_2 and b_3 . These parameters could, in principle, also change as a function of time, although they may be limited by constraints on secretory capacity per unit biomass. Changes in production due to total cell mass changes have been well documented in various endocrine systems, including the thyroid, the parathyroid, and pancreatic beta cells (Studer & Derwahl, 1995; Wada *et al*, 1997; Porat *et al*, 2011). Cell mass can grow by increasing cell numbers (hyperplasia) or by increasing cell size (hypertrophy). In the case of the HPA axis, both mechanisms occur – mice that were implanted with CRH-secreting tumors showed both corticotroph hyperplasia

and hypertrophy (Asa *et al*, 1992), and chronically stressed rats showed both adrenal hyperplasia and hypertrophy (Ulrich-Lai *et al*, 2006). The relative importance of each mechanism may be species-specific. A recent study on the growth of pancreatic acinar cells showed that mice relied mostly on hypertrophy while humans relied on hyperplasia, and that species differences in acinar cell size correlate with lifespan (Anzi *et al*, 2018). Characterizing relative contributions of hyperplasia and hypertrophy may have important implications for different aspects of the circuit, such as cell mass turnover rates, and rates of mutation accumulation in the tissues.

(3) Fig. 2b: when showing the cell masses, it may be useful to show the control cell masses somehow as well (preferably in gray). Also, label those rows "Cell masses".

We fixed this according to the suggestions of the reviewer.

(4) The cortisol baseline depends on very few parameters - but still depends on k -s and w -s. Also, this is the only hormone for which the baseline does not return to normal during prolonged stress. Why? In principle, could there be regulatory mechanisms to also push cortisol levels back to baseline? Why is it OK for human physiology to have high cortisol, but not OK to have abnormal levels of the other hormones?

We thank the reviewer for this comment, which made us revise the paragraph in the results which analyzes the compensation properties of cortisol. We discuss this now in the revised paragraph (page 15):

Cortisol in the present analysis is the only HPA hormone that does not return to baseline after prolonged stress input. This may correspond to its physiological role as a stress response hormone, with multiple physiological end-points, including modulation of immunity, metabolism, and behavior (McEwen, 1998). Cortisol therefore adjusts these end-points according to averaged stress levels, allowing ongoing response as long as stress persists. Together with the persistent responsiveness of cortisol to stress input u , the circuit allows cortisol level to be independent on almost all other parameters, including the hormone production rates per unit biomass (b_2, b_3) or removal rates (w_2, w_3), as well as the rates of the proliferation and removal of the adrenal cortex cells, k_A and w_A . This robustness is due to the ability of the functional masses to grow and shrink to buffer changes in these parameters. It allows cortisol to respond precisely to chronic stress input despite physiological variations in many of the circuit parameters.

(5) What would happen if both cell masses would depend on the same single hormone (either only x_1 or only x_2)?

To address this intriguing question, we now added a new appendix section, where we scanned all possible network topologies to see which of them have the properties observed in Figure 1. We find that only a small number of possible topologies show all of the observed features, and among them is the actual HPA circuit analyzed in this paper.

Appendix Section 3. Possible network topologies for the regulation of corticotroph and adrenal masses.

In this section we compare putative alternative topologies (interaction network architectures) for the regulation of tissue mass in the HPA axis. The experimentally observed interactions are the regulation of the pituitary corticotroph mass by CRH and the regulation of the adrenal cortex mass by ACTH. One could imagine other interactions between the hormones and gland masses. We were specifically interested in asking which topologies provide the features of the HPA hormones discussed here: exact adaptation of x_2 , hypercortisolemia during elevated stress for x_3 , and the dysregulation phenomena shown in Figure 1.

For this purpose, we consider all possible interactions between hormones and glands, where CRH, ACTH, and cortisol, can activate either the production or removal of cell mass of either the pituitary corticotrophs or the cortisol-secreting adrenal cortex cells. We describe this by the general equations:

$$\frac{dC}{dt} = w_C C (x_1^{a_{1C}} x_2^{a_{2C}} x_3^{a_{3C}} - x_1^{b_{1C}} x_2^{b_{2C}} x_3^{b_{3C}}) \quad [1]$$

$$\frac{dA}{dt} = w_A A (x_1^{a_{1A}} x_2^{a_{2A}} x_3^{a_{3A}} - x_1^{b_{1A}} x_2^{b_{2A}} x_3^{b_{3A}}) \quad [2]$$

where all of the powers have values of 0 or 1. The 12 powers give a total of $2^{12} = 4096$ circuits. Since the adrenal cortex cannot sense hypothalamic CRH which is secreted into a private portal vein system directly to the pituitary, we set $a_{1A} = b_{1A} = 0$. This results in $2^{10} = 1024$ circuits. We solved the fixed point analytically, to find that only 464 have a fixed point, and 111 of these have a stable positive fixed point. Of these, we excluded the circuits where $a_{iC} = b_{iC}$ or $a_{iA} = b_{iA}$, since these do not affect the sign of Eq. 1, 2 and therefore do not affect whether or not they have the mismatch property. Finally, only 6 circuits show exact adaptation for x_2 , as well as hypercortisolemia at high levels of input u . For all 6 circuits, Eq. 2 is $\frac{dA}{dt} = w_A A (x_2 - 1)$. Eq. 1 can be one of the following: (i) $\frac{dC}{dt} = w_C C (x_1 - 1)$, (ii) $\frac{dC}{dt} = w_C C (x_1 - x_2)$, (iii) $\frac{dC}{dt} = w_C C (x_1 - x_3)$, (iv) $\frac{dC}{dt} = w_C C (x_1 - x_2 x_3)$, (v) $\frac{dC}{dt} = w_C C (x_1 x_2 - 1)$, (vi) $\frac{dC}{dt} = w_C C (x_1 x_2 - x_3)$. Of these circuits only two circuits show the mismatch property after prolonged stress. These are circuit (i) which represents the experimentally observed interactions, and circuit (v) in which x_2 participates with x_1 to control corticotroph mass, in an ultrashort feedback loop. We conclude that the observed mass control interactions in the HPA axis are among the very few options that capture the observed phenomena.

We also refer to this SI section in the main text in the results (page 14).

“We also tested other putative circuit designs in which the hormones control mass changes in alternative ways, and find that the interactions considered here are among the very few designs that provide the observed HPA behavior (Appendix Section 3).”

Reviewer #3:

This is a very well written manuscript that suggests an explanation for data showing disconnection between ACTH secretion and CORT secretion following chronic stress. The beauty of the paper is the simplicity of the hypothesis: that everything can be explained by dynamic changes in the mass of the hormone-secreting cells. As demonstrated, if these changes occur on different slow time scales, then other features of the network (particularly negative feedback of cortisol) ensure that observations on hormone secretion levels that have been reported experimentally under a variety of stress-related conditions are replicated. No sensitivity analysis is needed, this is a very robust network response if one incorporates slow dynamic changes in cell mass. The work is a nice example of how a simple mathematical model can be used to explain ubiquitous data. I have only a few comments:

We thank the reviewer for this endorsement.

1) AVP is never mentioned. This is strange, since AVP is co-released with CRH from hypothalamic neurons and has a strong effect on corticotroph activity. I realize that much of the paper focusses on data from CRH tests (which don't involve AVP), but the physiological stimulatory role of AVP should at least be mentioned when the HPA axis is discussed.

We agree with the reviewer, and we now addressed this comment by adding a new paragraph in the discussion which discusses the role of AVP in the context of our model (page 16):

Another potentially important extension of the model will be to incorporate hypothalamic neuropeptides that modulate the secretion of ACTH, such as arginine vasopressin (AVP). AVP is released in response to various stressors and acts synergistically with CRH to promote ACTH secretion. AVP receptors in the pituitary are upregulated in response to high cortisol levels (Aguilera & Rabadan-Diehl, 2000), which may help maintain responsiveness of ACTH secretion to some stimuli during chronic stress. Another relevant neuropeptide is oxytocin, which inhibits ACTH secretion (Legros, 2001). These neuropeptides, which have a half-life of a few minutes, are important for determining the acute response of the HPA axis to input stimuli. Therefore, modelling changes in

their regulation on the slow timescale will help us better understand dysregulation after chronic HPA axis activation.

2) *Back in 2000 a model was published by Topp et al. (J. Theoretical Biology, 206:605) that demonstrated how beta-cell mass could change dynamically in response to insulin resistance so as to maintain glucose homeostasis. A later, and more biologically accurate, model was recently published by Ha et al. (Endocrinology, 157:624, 2016) that could explain other data including pathways to diabetes. Again, a main focus was on dynamic changes in the mass of the hormone-secreting (insulin in this case) cells. I get the feeling that the authors don't know about this work, which is based on dynamic compensation of cell mass (though I am not sure if that language was used). The authors only cite one of their own papers on the insulin-glucose system, which is also from 2016. Clearly, these other papers are relevant and should be cited and discussed as earlier models exhibiting dynamic compensation in an endocrine system.*

We thank the reviewer for bringing this to our attention. These references were in fact the basis of our own work on the glucose-insulin system, and we now added them to the introduction. We also added a new paragraph to the discussion which discusses the references, in the context of disease processes in the HPA axis (page 18):

The model presented in this study extends the literature on the feedback circuits that control the mass of endocrine organs. Such models are important for understanding disease processes that result in dysregulation of tissue mass, such as Type-2 Diabetes, in which insulin secretion by pancreatic β -cells becomes deficient. Pioneering work by Topp et al (Topp *et al*, 2000; De Gaetano *et al*, 2008) showed that β -cell death at high levels of glucose, known as glucotoxicity, can result in bistability, leading to long-term glucose dysregulation. This work was extended by a more biologically accurate model, that could explain additional data including pathways to diabetes (Ha *et al*, 2016). This bistability may be a side effect of a protective mechanism, because glucotoxicity can eliminate mutant β -cells that mis-sense glucose levels and as a result hyper-proliferate and hyper-secrete insulin (Karin & Alon, 2017). Another possible protective mechanism is autoimmune surveillance, where auto-reactive immune cells eliminate mis-sensing mutants (Kohanim *et al*, 2019), at the risk of the development of autoimmune diseases like Type-1 Diabetes. These mechanisms may be relevant for understanding certain diseases of the HPA glands, such as Cushing's disease due to ACTH-secreting pituitary adenomas, or Addison's disease due to autoimmune adrenal destruction (Kohanim *et al*, 2019).

3) *The Fig. 1 caption incorrectly states that the thicker curves are control.*

Fixed.

7th Jun 2020

Manuscript Number: MSB-20-9510R

Title: A new model for the HPA axis explains dysregulation of stress hormones on the timescale of weeks

Author: Omer Karin

Moriya Raz

Avichai Tandler

Alon Bar

Yael Korem Kohanim

Tomer Milo

Uri Alon

Thank you for sending us your revised manuscript. We have now heard back from the two reviewers who were asked to evaluate your study. As you will see the reviewers are satisfied with the modifications made and think that the study is now suitable for publication.

Before we can formally accept your manuscript, we would ask you to address a few remaining editorial issues listed below:

1. Please provide 5 keywords and incorporate them in the main text.
2. A Conflict of Interest statement should be provided in the main text.
3. Please remove the figures from the manuscript file. Figure legends need to be moved to the end of the manuscript.
4. Figures should be numbered in the order of their appearance in the text with Arabic numerals. Currently, Figure 3A is out of order and called out almost at the end of the manuscript. Please fix. Callouts to Figures S1&S2 needs to have the word "Appendix" added as a prefix- ie: Appendix Figure S1.
5. Our data editor has made a couple of suggestions on your manuscript (see attached), please address these issues.
6. Thank you for providing the synopsis image. However, the resolution of the text in the current image is a bit low. Can you provide a resized image (jpg format, 550px width and ~400px height) with enhanced resolution?
7. Please remove the Standfirst from the manuscript file. I have slightly shortened it, since the standfirst text cannot exceed 250 characters including space. Please let me know if you are fine with it or if you would like to introduce further modifications.

When you resubmit your manuscript, please download our CHECKLIST (<http://bit.ly/EMBOPressAuthorChecklist>) and include the completed form in your submission. *Please note* that the Author Checklist will be published alongside the paper as part of the transparent process (<https://www.embopress.org/page/journal/17444292/authorguide#transparentprocess>)

Click on the link below to submit your revised paper.

I look forward to receiving your revised manuscript as soon as possible.

Sincerely,
Jingyi

Jingyi Hou
Editor
Molecular Systems Biology

If you do choose to resubmit, please click on the link below to submit the revision online before 7th Jul 2020.

IMPORTANT: When you send your revision, we will require the following items:

1. the manuscript text in LaTeX, RTF or MS Word format
2. a letter with a detailed description of the changes made in response to the referees. Please specify clearly the exact places in the text (pages and paragraphs) where each change has been made in response to each specific comment given
3. three to four 'bullet points' highlighting the main findings of your study
4. a short 'blurb' text summarizing in two sentences the study (max. 250 characters)
5. a 'thumbnail image' (550px width and max 400px height, Illustrator, PowerPoint or jpeg format),

which can be used as 'visual title' for the synopsis section of your paper.

6. Please include an author contributions statement after the Acknowledgements section (see <https://www.embopress.org/page/journal/17444292/authorguide#manuscriptpreparation>)

7. Please complete the CHECKLIST available at (<http://bit.ly/EMBOPressAuthorChecklist>). Please note that the Author Checklist will be published alongside the paper as part of the transparent process

(<https://www.embopress.org/page/journal/17444292/authorguide#transparentprocess>).

8. Please note that corresponding authors are required to supply an ORCID ID for their name upon submission of a revised manuscript (EMBO Press signed a joint statement to encourage ORCID adoption) (<https://www.embopress.org/page/journal/17444292/authorguide#editorialprocess>).

Currently, our records indicate that the ORCID for your account is 0000-0003-1121-5907.

Link Not Available

The system will prompt you to fill in your funding and payment information. This will allow Wiley to send you a quote for the article processing charge (APC) in case of acceptance. This quote takes into account any reduction or fee waivers that you may be eligible for. Authors do not need to pay any fees before their manuscript is accepted and transferred to the publisher.

*** PLEASE NOTE *** As part of the EMBO Press transparent editorial process initiative (see our Editorial at <http://dx.doi.org/10.1038/msb.2010.72> , Molecular Systems Biology will publish online a Review Process File to accompany accepted manuscripts. When preparing your letter of response, please be aware that in the event of acceptance, your cover letter/point-by-point document will be included as part of this File, which will be available to the scientific community. More information about this initiative is available in our Instructions to Authors. If you have any questions about this initiative, please contact the editorial office (msb@embo.org).

REFeree REPORTS

Reviewer #1:

I would like to thank the Authors for addressing my questions. I would like to recommend that this interesting manuscript is accepted for publication.

Reviewer #3:

The authors have done a nice job with the revision. I have no additional suggestions for modification.

Corresponding Author Name: Uri Alon

Manuscript Number: MSB-20-9510